# Active Learning Methodology for Expert-Assisted Anomaly Detection in Mobile Communications

**DOI:** 10.3390/s23010126

**Published:** 2022-12-23

**Authors:** José Antonio Trujillo, Isabel de-la-Bandera, Jesús Burgueño, David Palacios, Eduardo Baena, Raquel Barco

**Affiliations:** 1Instituto de Telecomunicación (TELMA), Universidad de Málaga, CEI Andalucía TECH E.T.S. Ingeniería de Telecomunicación, Bulevar Louis Pasteur 35, 29010 Málaga, Spain; 2Tupl Spain, Tupl Inc., Campus de Teatinos, 29071 Málaga, Spain

**Keywords:** active learning, anomaly detection, 5G, machine learning, self-organizing networks

## Abstract

Due to the great complexity, heterogeneity, and variety of services, anomaly detection is becoming an increasingly important challenge in the operation of new generations of mobile communications. In many cases, the underlying relationships between the multiplicity of parameters and factors that can cause anomalous behavior are only determined by human expert knowledge. On the other hand, although automatic algorithms have a great capacity to process multiple sources of information, they are not always able to correctly signal such abnormalities. In this sense, this paper proposes the integration of both components in a framework based on Active Learning that enables enhanced performance in anomaly detection tasks. A series of tests have been conducted using an online anomaly detection algorithm comparing the proposed solution with a method based on the algorithm output alone. The obtained results demonstrate that a hybrid anomaly detection model that automates part of the process and includes the knowledge of an expert following the described methodology yields increased performance.

## 1. Introduction

The management of mobile networks is becoming increasingly difficult and costly due to the high complexity of the new generations of the standard. To overcome this, network management automation becomes a key factor, of even more relevance in 5G networks [1], where new services and scenarios [2] complicate their correct operation even more. Such automation is achieved through the implementation of SON (Self Organizing Networks) [3], which enables configuration, optimization and healing tasks, among others, through the use of algorithms.

As part of these tasks, fault detection and healing become critical due to the negative consequences that possible issues can cause on the network performance and the perceptible QoE (Quality of Experience) for the user. Manual detection of these network faults is a time-consuming process due to the large data size, leading to the urge for automation of anomaly detection in mobile networks. However, this automation is often problematic, as it is difficult for the algorithms to detect these strange behaviors in the data obtained from the network.

For this reason, a viable alternative to automate this process could be a mixed solution: the use of an automated algorithm and a hands-on approach to make the process less costly in terms of time and resources while leveraging the insight of a network management expert.

Within the field of machine learning, the concept of active learning is defined as a methodology based on including the opinion of a human expert [4] to improve the performance of automated algorithms. Consequently, the advantages obtained from a machine learning algorithm are combined in the same approach with the knowledge that an expert in network management could provide. Therefore, knowledge acquisition becomes a critical element in any active learning scheme, being able to perform more generic functions as in [5], or to focus on building a model for a specific task as proposed in [6], where a model for diagnosis is implemented from the acquired knowledge. In [7], the authors propose a hybrid framework where knowledge acquisition is included as part of the troubleshooting process.

This paper presents a framework based on the concept of active learning for anomaly detection in mobile networks. Therefore, the objective is not to propose a new detection algorithm but to improve the performance of an existing one by using the knowledge of a network management expert. The main contributions of this work are:A novel methodology capable of adapting to every anomaly detection algorithm and improving its performance by using the knowledge of experts;An active learning technique for anomaly detection adapted to a complex environment such as mobile networks;A real-time analysis capability, which allows changes to be made if improvements are not achieved;A user-friendly interface is included so that the expert can provide feedback.

The rest of this paper is organized as follows: Section 2 presents an introduction to the related work in the literature. Section 3 presents the methodology proposed. In Section 4, a use case with a concrete detection algorithm is implemented. Then, in Section 5, the tests and results used to validate the method are presented. Finally, Section 6 concludes the paper.

## 2. Related Work

This section summarizes related studies from two aspects: anomaly detection and active learning in cellular networks.

### 2.1. Anomaly Detection

Anomaly detection is a procedure for finding data that differs from the dataset’s expected behavior or pattern [8]. Traditionally, many techniques and methods have been applied to automate this task, as reported by the authors of [9], who classify them into statistical, learning-based, and information-theoretic. In addition, a new vision for anomaly detection based on big data are proposed, which the authors call DNA (Deep Network Analyzer). In the same field based on the application of big data is the solution presented in [10]. In particular, the system recommends dividing different city areas according to their similar traffic pattern by analyzing a large amount of data collected from the network. Then, the k-means algorithm is applied to each zone, allowing the classification between normal behavior and anomalies. This same k-means technique is employed by the authors of [11], who propose a framework for anomaly detection and traffic prediction in mobile networks.

In addition, statistical methods are a simpler alternative to big data, as it is not necessary to analyze all the data. Instead, they are based on the calculation of statistics on this data, as, for example, proposed in [12], where the mean and standard deviation are used as the basis for anomaly detection. Similarly, the authors of [13] also use some statistics combined with a set of thresholds that are used to compare and decide whether the analyzed data are abnormal.

Other more elaborate proposals that use several mechanisms and techniques in the same method for anomaly detection, as in [14]. The authors propose a procedure in which the first decomposition of the time series is performed. Then, several artificial intelligence algorithms such as SARIMA (Seasonal Autoregressive Integrated moving average) and MSCRED (Multi-Scale Convolutional Recurrent Encoder-Decoder) are applied. Moreover, in [15], a hybrid system based on several machine learning techniques is proposed. More precisely, techniques such as OCSVM (One Class Support Vector Machine), SVR (Support Vector Regression), or LSTM (Long Short Term Memory) are used for the detection of spatio-temporal anomalies in mobile networks.

### 2.2. Active Learning

Active learning is a technique that has aroused great interest, as shown in the literature, becoming one of the most suitable alternatives to automate the task of anomaly detection. In [16], an active learning framework with the aim of detecting rare-category data and anomalies is proposed. Initially, a semi-supervised algorithm is improved with the expert feedback until the algorithm evolves to an unsupervised mode. On the other hand, the reduction of human effort is a key factor in the method proposed in [17], where an expert only labels a subset of environmental data to train different supervised anomaly detection algorithms. A similar concept is followed in [18], with a system to transform any deep learning detection algorithm into an active learning model through expert feedback, including knowledge acquisition.

Other works suggest the application of classical anomaly detection for time series, combined with the use of active learning. An illustrative example is presented in [19], where this technique is employed with several classifiers to detect anomalies in the electrocardiogram time series. Following related lines of research, an improved reinforcement learning algorithm for anomaly detection with active learning is presented in [20]. This approach takes advantage of expert feedback to label the samples and improve the performance of the algorithm. However, none of these works are focused on such sophisticated environments as a mobile network, where the data and anomalies encountered can be very diverse.

## 3. Proposed Methodology

This section describes the operation of the proposed methodology, illustrated in Figure 1 as a block diagram. In order to apply an active learning methodology, the present work aims to use feedback from an expert on the anomalies detected by an automatic algorithm. More specifically, the expert assigns a label, indicating whether the data indeed correspond to abnormal behavior. Subsequently, this knowledge gained from the expert is stored in a database. Later, the acquired insight is used to fine-tune the detection algorithm and thus improve its performance. Depending on the detection algorithm employed, different modifications can be made, such as adjusting the hyperparameters or retraining with newly labeled samples, among others.

In Figure 1, several types of blocks can be identified. First, there is the database mentioned above to store the acquired knowledge of the expert. Then, there are the blocks that perform a connection function between the detection algorithm and the rest of the system, which are highlighted in orange. Finally, the blocks operating independently of the selected detection algorithm are shaded in green.

Each block in the diagram shown in Figure 1 performs a specific function:Knowledge Acquisition. Block that gathers the knowledge of the expert about the various anomalies detected by the algorithm. In this way, each anomaly presented is categorized by the expert with one of the following labels:-Confirmed. It indicates that the expert agrees with the anomaly detected by the algorithm and qualifies it as abnormal data;-Dismissed. It indicates that the expert considers that the anomaly presented by the algorithm is not such and rejects it;-Modified. The expert modifies the beginning or end of the anomalous interval presented by the algorithm, thus modifying the anomaly;-Added. The expert can add an anomaly not detected by the algorithm;Feedback to Database. This stage is responsible for keeping the database updated, adding new comments from the expert at each iteration, and avoiding formatting errors or duplicities;Results. The block presents the data of the anomalies detected by the algorithm in a user-friendly way to the expert;Interface I. Depending on the selected detection algorithm, which must adapt the knowledge acquired and stored in the database to improve the performance of the detection algorithm;Interface II. It formats the most relevant output information of the detection algorithm. As in Interface I, the implementation of this block depends on the data provided at the output by the selected detection algorithm;Detection Algorithm. It represents the detection algorithm that has been selected to find anomalies in the analyzed data;Database block. Block in charge of storing all anomalies already analyzed and labeled by an expert and keeping them available so that other blocks may use this information.

At each iteration of the loop, new samples are analyzed by the detection algorithm, and their results are presented to the expert in order to gain more knowledge and achieve the best possible performance of the detection algorithm. Therefore, the operating loop of active learning in the proposed methodology is as follows:1.Initial training of the detection algorithm if needed;2.Execution of the detection algorithm with the latest data collected in the network (Detection Algorithm);3.Extraction of useful information on detected anomalies (Interface II);4.User-friendly reporting of anomalies to the expert (Results);5.Collection of the feedback from the expert on the anomalies reported (Knowledge Acquisition);6.Update of the database with the latest reviewed data (Feedback to Database);7.Adaptation and changes in the detection algorithm based on the knowledge available in the database (Interface I);8.Back to step 2.

After each iteration of the algorithm, the results are measured to see if the algorithm’s effectiveness improves or worsens so that appropriate decision-making can be undertaken.

## 4. Use Case

This section describes a practical implementation of the method proposed in the preceding section. A threshold-based statistical method proposed in [13] has been chosen as the detection algorithm. As described in the previous section, the Interface I and II blocks need to be implemented according to the selected detection algorithm.

### 4.1. Anomaly Detection Algorithm

The algorithm selected for this use case is described in [13], which is based on comparing the latest KPI (Key Performance Indicator) data with the most recent behavior in the network with the aim of identifying deviations and detecting anomalies. Using the mean and standard deviation of past samples as reference, current KPI values are analyzed, and if they significantly differ from recent behavior, an alarm is generated. A set of thresholds (low, medium, and high) is used to differentiate the magnitude of the alarms. The temporal concatenation of several alarms within a few hours is regarded as an anomaly by this algorithm. In view of this, the proposed methodology will modify the aforementioned thresholds in order to try to obtain the best possible algorithm performance.

### 4.2. Results

First of all, the Results block represents the anomalies in a simple way, with the help of the graphs presented in Figure 2. Specifically, Figure 2a shows the history of the data analyzed, highlighting the intervals with anomalies detected by the algorithm. Figure 2b shows the anomalous intervals since the last execution of the active learning loop proposed in the methodology.

### 4.3. Knowledge Acquisition

The knowledge acquisition block allows the expert to confirm or discard the reported anomalies, as well as to modify the interval over which the anomalies extend or to add an interval not detected by the algorithm. In this way, the anomaly data are assigned a label and stored in the database.

For this purpose, a friendly interface is displayed to the expert along with anomalies detected in the form of graphs such as Figure 2. This interface is similar to the one shown in Figure 3, where a table with the most important information about the detected anomaly is shown. There are two buttons to confirm/dismiss the anomaly, another to modify the dates, and one to add a new anomaly not detected in the displayed data. Finally, there is a button to send all the information to the next block.

### 4.4. Feedback to Database

The Feedback to Database block includes the latest intervals reviewed by the expert and keeps the database up to date and error-free. Therefore, anomalous intervals detected by the algorithm and reviewed by the expert (labeled as confirmed, dismissed, modified, or added) are kept in the knowledge database.

### 4.5. Interface I

This interface oversees the application of the stored knowledge to the detection algorithm, with the aim of reaching an optimal performance. In this case, the algorithm used in this case applies different thresholds to determine the level of anomaly of each sample separately, and accordingly detects the anomalous intervals within the data.

In this use case, as described above, the algorithm uses a set of three thresholds, which will be modified based on the knowledge acquired until optimal performance is reached. To adapt the acquired knowledge, an update factor ($f_{\mathrm{update}}$) is calculated based on the labels of each of the database records (confirmed, dismissed, modified, and added). The computation of such factor is made following the Equation (Equation 1):(1)$$f_{\mathrm{update}}=\frac{\mathrm{Confirmed}-\mathrm{Dismissed}+\mathrm{Added}}{\mathrm{Total}}$$
where Confirmed, Dismissed, and Added are the number of entries in the database with that label, and Total is the number of available entries. In this way, it is possible to modify the thresholds, reinforcing the decision of the algorithm when the expert agrees with the decision or penalizing it when the opinion of the expert is different from the one provided by the algorithm.

Once this factor has been obtained, the thresholds are updated according to (Equation 2):(2)$${\mathrm{Threshold}}_{\mathrm{new}}={\mathrm{Threshold}}_{\mathrm{old}}+{\mathrm{Threshold}}_{\mathrm{old}}*f_{\mathrm{update}}$$
where ${\mathrm{Threshold}}_{\mathrm{old}}$ is the threshold in the last iteration, and ${\mathrm{Threshold}}_{\mathrm{new}}$ is the new value for the next iteration of the loop. The way in which the threshold is updated allows it to not increase or decrease its values abruptly, thus finding the optimum point without overlooking it.

### 4.6. Interface II

In this case, interface II is responsible for formatting and extracting useful information from each anomalous interval detected by the algorithm. Specifically, the following information is obtained for each interval: start and end dates of the anomaly, anomalous KPI, mean value if the behavior was normal, and mean value observed in the interval. These data are displayed in the interface of Figure 3, which the expert uses to express his opinion about the anomalies. In addition, to facilitate the graphical representation of the anomalies, the values of each KPI analyzed in the network are also extracted, both historically and since the last iteration, as well as the usual behavior. Instead, these data and graphs are used by the results block, as in Figure 2 to show more simply the anomalies to the expert.

## 5. Experiments and Results

The performance of the proposed methodology has been validated by performing a series of tests on a dataset obtained from commercial LTE networks. Furthermore, comparative analysis is carried out against the results obtained by the detection algorithm used in [13] without the application of the methodology described in this work. The available data cover a total of 13 cells, where six KPIs are investigated. For each KPI, the total number of samples collected during 25 working days, on an hourly basis, gives a total of 576 samples. Five of these days (i.e., 120 samples) are used for the initial training of the algorithm, and the rest have been assigned to test the efficiency of the method.

As mentioned above, in each iteration, changes in the thresholds of the detection algorithm are executed. Therefore, the optimal thresholds of [13] are established as target values, and the initial thresholds are adjusted to different values so that the detection algorithm is not optimized. Hence, the goal is to reach values similar to the optimal ones, as shown in Table 1.

First, the evolution of the knowledge acquired by the database has been analyzed by observing the progression of the number of anomalous intervals for each label. Figure 4 reveals that, after several iterations, the number of discarded or added samples does not increase, while the number of confirmed samples rises considerably. Such a fact demonstrates that the algorithm has learned from the expert’s knowledge to detect anomalies successfully, increasing its efficiency and improving its performance.

Secondly, it is observed how the proposed methodology fine-tunes the values of the thresholds according to the knowledge acquired by the expert as shown in Figure 5. The trend of these thresholds illustrates how the values are approaching the target values described above, thus reaching an optimal configuration of the detection algorithm.

Finally, the results have been statistically analyzed using the same statistics that [13]. For this purpose, the following counters are calculated:True Positive (*TP*). Number of abnormal labeled samples which are correctly detected as abnormal by the methodology;True Negative (*TN*). Number of non-abnormal labeled samples which are correctly detected as non-abnormal by the methodology;False Positive (*FP*). Number of non-abnormal labeled samples which are wrongly detected as abnormal by the methodology;False Negative (*FN*). Number of abnormal samples which are wrongly classified as non-abnormal by the methodology.

Using these counters, the following metrics can be calculated, which will provide a general overview of the performance of the proposed methodology.

Accuracy: correctly detected samples of the total per day. It is calculated with Equation (Equation 3):
(3)$$\mathrm{Accuracy}=\frac{\mathrm{TP}+\mathrm{TN}}{\mathrm{TP}+\mathrm{TN}+\mathrm{FP}+\mathrm{FN}}$$Error Rate: wrongly detected samples of the total per day. It is calculated with Equation (Equation 4):
(4)$$\mathrm{Error}\;\mathrm{Rate}=\frac{\mathrm{FP}+\mathrm{FN}}{\mathrm{TP}+\mathrm{TN}+\mathrm{FP}+\mathrm{FN}}=1-\mathrm{Accuracy}$$False Positive Rate (FPR): wrongly detected samples of the non-anomalous samples per day. It is calculated with Equation (Equation 5):
(5)$$\mathrm{FPR}=\frac{\mathrm{FP}}{\mathrm{TN}+\mathrm{FP}}$$Precision: correctly detected samples of the anomalous labeled samples per day. It is calculated with Equation (Equation 6):
(6)$$\mathrm{Precision}=\frac{\mathrm{TP}}{\mathrm{TP}+\mathrm{FP}}$$Recall: correctly detected samples of the all anomalous labeled samples per day. It is calculated with Equation (Equation 7):
(7)$$\mathrm{Recall}=\frac{\mathrm{TP}}{\mathrm{TP}+\mathrm{FN}}$$

Figure 6 depicts the obtained enhanced performance in statistical terms. The accuracy achieved by the algorithm is close to 100% after an initial period that is coincident with the period in which the most changes are made to thresholds. On the other hand, the Error Rate and FPR values remain at minimum levels after this period of threshold readjustment. In this sense, the values of Recall and Accuracy also reach 100%, achieving remarkable results and leading to an almost perfect optimization of the thresholds of the detection algorithm and, therefore, of its performance.

Finally, Table 2 summarizes the values obtained for the different metrics concerning both the proposed methodology and the anomaly detection algorithm without considering the expert’s feedback. It is noteworthy that the values shown for the first case have been taken after the threshold equalization period.

## 6. Conclusions

In this work, a new assisted anomaly detection methodology based on the concept of active learning has been proposed.

Given the difficulty in anomaly detection during mobile network operation, this proposal can reduce the amount of work of an expert in such a task, achieving equal or even better results than fully manual management. Furthermore, it allows the optimization of fully automatic detection methods that could enhance their performance with expert feedback.

Thus, the proposed methodology includes a user-friendly way for the expert to express his assessment of the detected anomalies. This avoids the time cost for the expert engineer of analyzing a large amount of data to detect anomalies since those that are detected by the algorithm are analyzed by simply looking at the anomaly graphs. More specifically, the implemented use case demonstrates how well the approach works. Thus, a proof of concept is carried out starting from non-optimized thresholds that are updated every 24 h based on the expert’s opinion.

The results indicate that, after only five days of application of the proposed methodology, the values are considered optimal for the selected detection algorithm. Similarly, despite taking five days to optimize the algorithm, better metrics are achieved than without applying the active learning methodology.

In addition, the proposed approach has proven to be well scalable, both because it can be adapted to almost any anomaly detection algorithm and because it is based on the study and analysis of KPIs. These metrics are most common in mobile networks, whether LTE, 5G, or those to come.

Further lines of research to be considered for this work arise in this field. First, the implementation of other use cases with different detection algorithms is proposed. A second line could include tests with data from other mobile technologies, earlier or later, although these should be previously tested on the detection algorithms to be applied. Finally, a less flexible version could be made that implements a proprietary detection algorithm that fits perfectly with the proposed methodology.

## Figures and Tables

**Figure 1 sensors-23-00126-f001:**
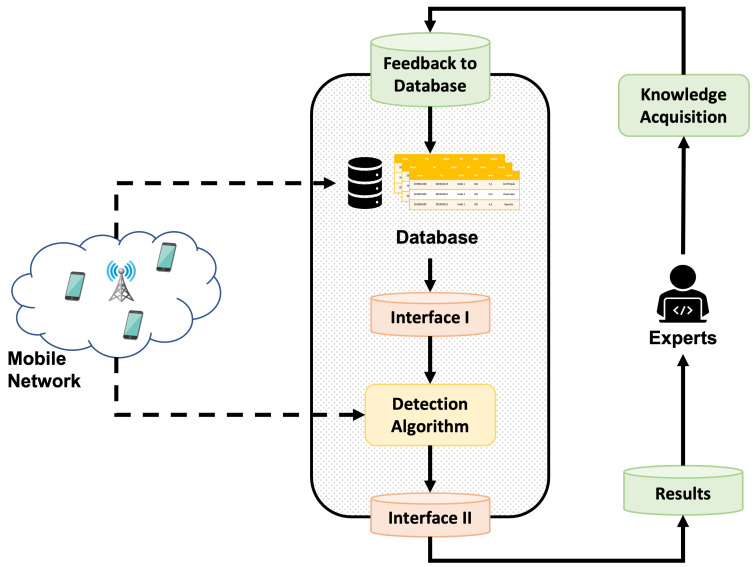
Blocks diagram of proposed methodology.

**Figure 2 sensors-23-00126-f002:**
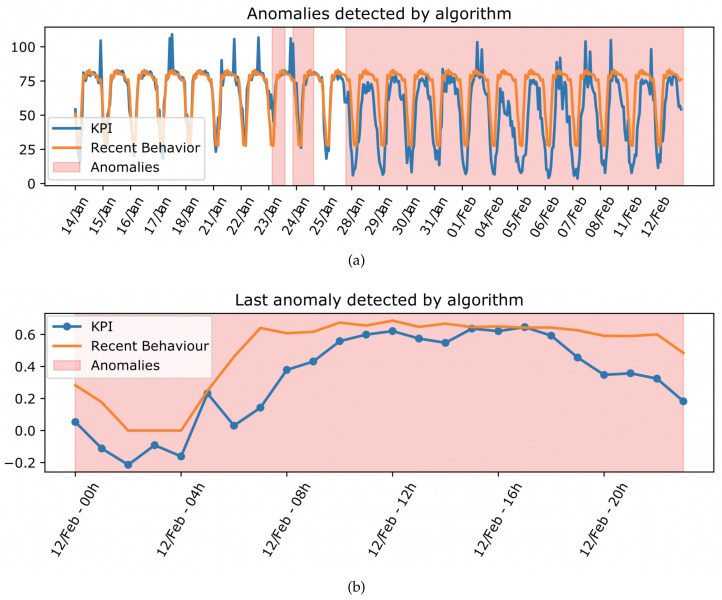
Example of graphics shown to an expert. (**a**) Historical data analyzed; (**b**) Last hours analyzed.

**Figure 3 sensors-23-00126-f003:**
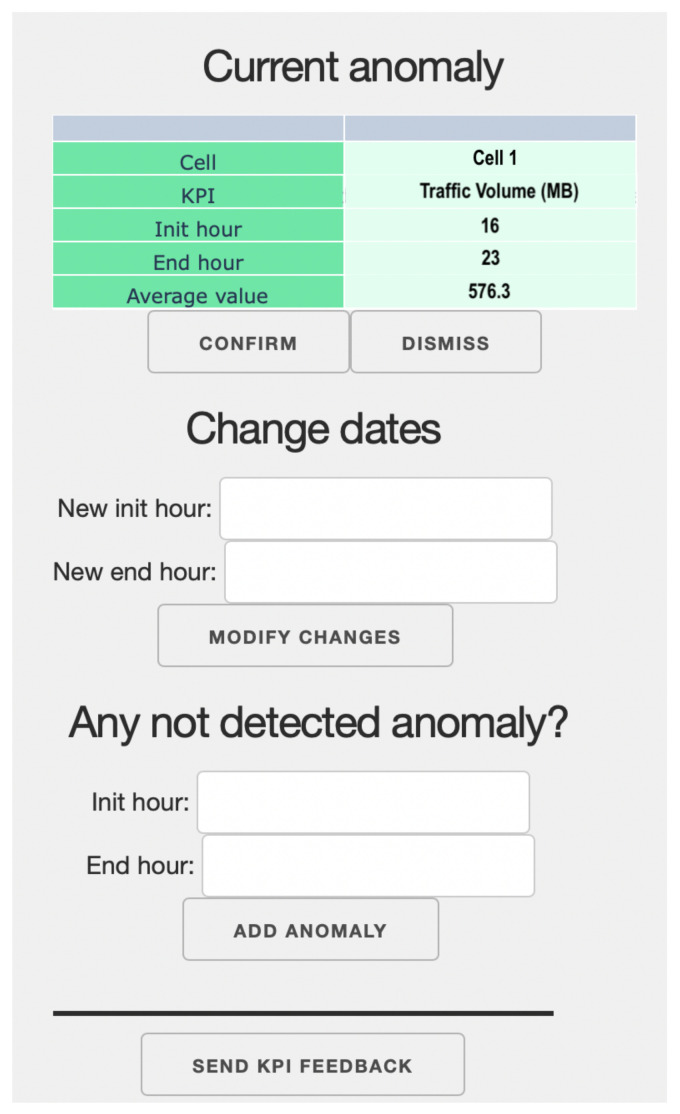
Example of graphics shown to an expert.

**Figure 4 sensors-23-00126-f004:**
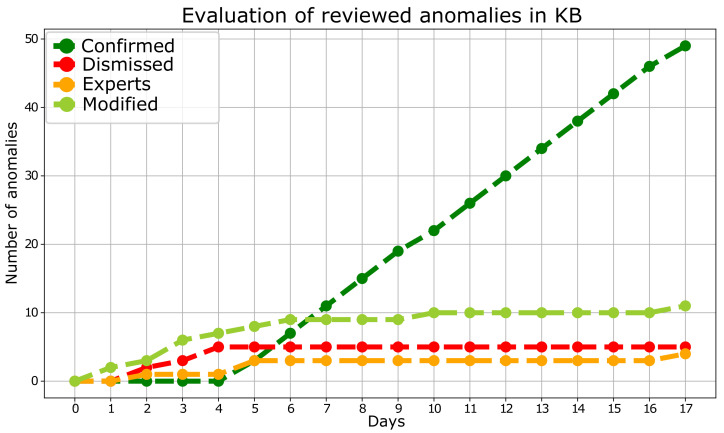
Evolution of the number of intervals of each case of anomaly.

**Figure 5 sensors-23-00126-f005:**
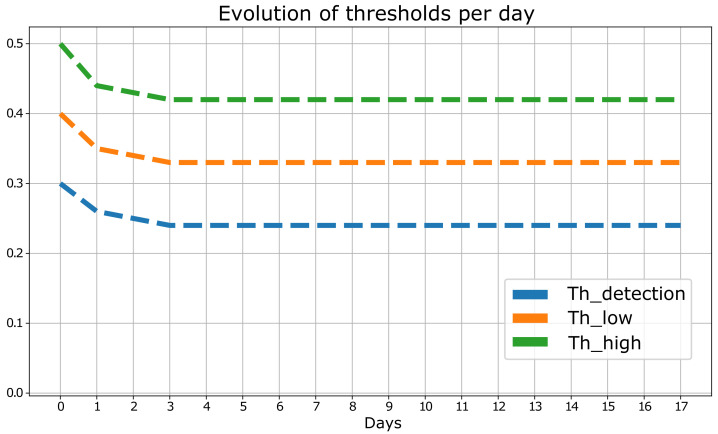
Evolution of detection algorithm thresholds.

**Figure 6 sensors-23-00126-f006:**
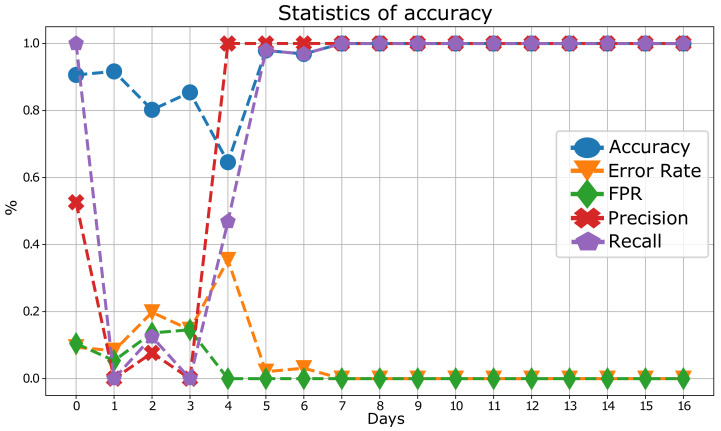
Statistics of algorithm using the proposed methodology.

**Table 1 sensors-23-00126-t001:** Initial vs. Target Thresholds.

Threshold	Initial	Target
Low	0.3	0.27
Medium	0.4	0.33
High	0.5	0.43

**Table 2 sensors-23-00126-t002:** Anomaly Detection vs. Active Learning methodology.

Metric	Only Anomaly Detection	AD with Active Learning Methodology
Accuracy	0.98	1.0
Error Rate	0.216	0.0
FPR	0.135	0.0
Precision	0.7	1.0
Recall	0.78	1.0

## Data Availability

Not applicable.

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
