# Peer review of "Active Learning Methodology for Expert-Assisted Anomaly Detection in Mobile Communications"

_sensors, 2022, doi:10.3390/s23010126_

Round 1
Reviewer 1 Report
There are multiple issues in the present document that must be fixed prior to acceptance. Several crucial components are lacking.. In addition, I find little uniqueness in their work. Before it can be considered acceptable, it needs extensive adjustment, in my opinion. Before it can be accepted, the following factors must be considered:
1- Related work is missing.
2- What is the proposed model? The current version of the manuscript is hard to follow.
3- What is the measurement used in the experiment?
4-The conclusion section should be rewritten again to enhance the work.
5- The abstract is weak and needs to be written again.
6- The reference needs to be outdated and updated.
7-No hyperparameters were used in the experiment.
8- In their experiment, no comparisons were made.
Reviewer 2 Report
This paper studies the anomaly detection of mobile communication, and the experimental results show that the expert's perspective based on active learning proposed in this paper significantly affects the anomaly detection of mobile communication. However, this paper has several limitations, and the standard is not enough. Addressing these following items would result in a good article.
(1) The abstract explains the research question, purpose and scheme but lacks the experimental scheme design and conclusion. If it can be supplemented, it will be more eye-catching.
(2) In the introduction, there is no introduction to the Anomaly detection methods of traditional mobile communication. The supplementary explanation on this aspect will make your article more complete.
(3) It is suggested to increase the diversity of training samples and compare the proposed method with the traditional detection method to verify the superiority of the proposed algorithm.
(4) Planning for future research is best mentioned in the article, which can guide readers or other researchers for related work.
(5) Pay attention to the uniform formatting of the paper. For example, the format of the references should be the same. Moreover, please review the article again and correct the minor errors.
Round 2
Reviewer 1 Report
The manuscript provided by the author solved all previous issues. However, the only edit is needed in the abstract.
Author Response
Following the reviewer's recommendation, some changes have been made in the abstract of the article that have been included in the new version of the article uploaded. Likewise, the new abstract with the changes in bold below:
"Due to the great complexity, heterogeneity, and variety of services, anomaly detection is becoming an increasingly important challenge in the operation of new generations of mobile communications. In many cases, the underlying relationships between the multiplicity of parameters and factors that can cause anomalous behavior are only known by human expert knowledge. On the other hand, although automatic algorithms have a great capacity to process multiple sources of information, they are not always able to correctly signal such abnormalities. In this sense, this paper proposes the integration of both components in a framework based on Active Learning that enables enhanced performance in anomaly detection tasks. A series of tests have been conducted using an online anomaly detection algorithm comparing the proposed solution with a method based on the algorithm output alone. The obtained results demonstrate that a hybrid anomaly detection model that automates part of the process and includes the knowledge of an expert following the described methodology yields increased performance."